# Noise-Induced Vascular Dysfunction, Oxidative Stress, and Inflammation Are Improved by Pharmacological Modulation of the NRF2/HO-1 Axis

**DOI:** 10.3390/antiox10040625

**Published:** 2021-04-19

**Authors:** Maria Teresa Bayo Jimenez, Katie Frenis, Swenja Kröller-Schön, Marin Kuntic, Paul Stamm, Miroslava Kvandová, Matthias Oelze, Huige Li, Sebastian Steven, Thomas Münzel, Andreas Daiber

**Affiliations:** 1Department of Cardiology, Cardiology I, University Medical Center of the Johannes Gutenberg-University, Langenbeckstraße 1, 55131 Mainz, Germany; maitebaji@gmail.com (M.T.B.J.); katiefrenis@gmail.com (K.F.); swenja.kroeller-schoen@gmx.de (S.K.-S.); marin.kuntic93@gmail.com (M.K.); PaulStamm@gmx.de (P.S.); miroslava.kvandova@gmail.com (M.K.); matzeoelze@aol.com (M.O.); tmuenzel@uni-mainz.de (T.M.); 2Department of Pharmacology, University Medical Center of the Johannes Gutenberg-University, Langenbeckstraße 1, 55131 Mainz, Germany; huigeli@uni-mainz.de; 3German Center for Cardiovascular Research (DZHK), Partner Site Rhine-Main, 55131 Mainz, Germany

**Keywords:** environmental risk factors, aircraft noise exposure, inflammation, endothelial dysfunction, oxidative stress, heme oxygenase-1, NRF2

## Abstract

Vascular oxidative stress, inflammation, and subsequent endothelial dysfunction are consequences of traditional cardiovascular risk factors, all of which contribute to cardiovascular disease. Environmental stressors, such as traffic noise and air pollution, may also facilitate the development and progression of cardiovascular and metabolic diseases. In our previous studies, we investigated the influence of aircraft noise exposure on molecular mechanisms, identifying oxidative stress and inflammation as central players in mediating vascular function. The present study investigates the role of heme oxygenase-1 (HO-1) as an antioxidant response preventing vascular consequences following exposure to aircraft noise. C57BL/6J mice were treated with the HO-1 inducer hemin (25 mg/kg i.p.) or the NRF2 activator dimethyl fumarate (DMF, 20 mg/kg p.o.). During therapy, the animals were exposed to noise at a maximum sound pressure level of 85 dB(A) and a mean sound pressure level of 72 dB(A). Our data showed a marked protective effect of both treatments on animals exposed to noise for 4 days by normalization of arterial hypertension and vascular dysfunction in the noise-exposed groups. We observed a partial normalization of noise-triggered oxidative stress and inflammation by hemin and DMF therapy, which was associated with HO-1 induction. The present study identifies possible new targets for the mitigation of the adverse health effects caused by environmental noise exposure. Since natural dietary constituents can achieve HO-1 and NRF2 induction, these pathways represent promising targets for preventive measures.

## 1. Introduction

Most traditional cardiovascular risk factors such as diabetes, smoking, and hyperlipidemia are associated with vascular complications such as endothelial dysfunction [1]. During the last decades, it has been shown that environmental stressors such as air pollution and excess of noise may facilitate the development of cardiovascular diseases (CVD) [2,3]. We previously identified oxidative stress and inflammation as central molecular mechanisms incurring vascular dysfunction following noise exposure [4,5]. A peak sound level of 85 db (A) and a mean sound level of 72 db (A) applied for 1, 2, and 4 days caused an increase in systolic blood pressure and circulating stress hormones, as well as oxidative stress and inflammation, resulting in endothelial dysfunction.

From a clinical perspective, because cardiovascular diseases are characterized by oxidative stress and inflammation, the actions of antioxidant enzymes could represent a new pharmacological strategy. Heme oxygenase-1 (HO-1) catalyzes the degradation of heme into ferrous iron (Fe^2+^), carbon monoxide (CO), and biliverdin, which is subsequently converted into bilirubin [6]. The principal physiological function of these catalytic products is the protection from oxidative stress and inflammation [7,8]. Accordingly, in some studies of CVD, the HO-1 activator hemin demonstrated beneficial therapeutic effects, such as normalization of vascular reactive oxygen species levels or improvement of vascular function [9,10,11]. In addition, nuclear-factor erythroid 2-related factor 2 (NRF2) regulates the expression of some human genes that are involved in the antioxidant response and participates in multiple homeostatic functions [12], including HO-1. It is for this reason that NRF2 activators, including dimethyl fumarate (DMF), have been utilized in studies demonstrating improvement in antioxidant response [13,14,15,16]. Therefore, we aimed to investigate the role of HO-1 induction or NRF2 activation in noise-induced vascular dysfunction, arterial hypertension, oxidative stress, and inflammation using treatment with hemin and DMF.

## 2. Materials and Methods

### 2.1. Animals

All animals were treated in accordance with the Guide for the Care and Use of Laboratory Animals as adopted by the US National Institutes of Health with approval granted by the Ethics Committee of the University Medical Center Mainz and the Landesuntersuchungsamt Rheinland-Pfalz (Koblenz, Germany; permit numbers 23 177-07/G 18-1-084 and 23 177-07/G 15-1-094). We used male C57BL/6J mice with an age between 6 and 12 weeks purchased from Janvier. The mice were housed in a standard 12 h light/dark cycle with free access to food and water. Mice were subjected to activator HO-1 treatment (hemin or DMF) and noise exposure for the subsequent 4 days. The different animal programs were conducted on different days with in-day controls and noise groups for each experiment.

### 2.2. Sacrifice of Animals, Organ Removal, and Sample Preparation

Animals were terminally anesthetized by isoflurane, killed by cervical dislocation, and dissected from the abdomen. The diaphragm and ribs were cut, and 200 μL of 1:6 heparin was injected into the beating heart to prevent clotting. Blood was collected via syringe and transferred into monovettes (Sarstedt, Nümbrecht, Germany) for centrifugation, and the plasma was removed and frozen for future measurements. Afterwards, heart, kidney, liver, aorta, and brain were carefully dissected and transferred to ice-cold Krebs-Hepes (KH) buffer (99 mM NaCl, 4.69 mM KCl, 2.5 mM CaCl_2_, 1.2 mM MgSO_4_, 25 mM NaHCO_3_, 1.03 mM K_2_HPO_4_, 20 mM Na-Hepes, 11 mM D-glucose; pH 7.35), cleaned, and portioned. All tissue samples were stored at −80 °C without solutions (dry). The fresh aorta was placed in a Petri dish with KH buffer; the vessels were carefully cleaned off adhesive adipose and connective tissue. Subsequently, the aortas were cut into rings (3–4 mm length) and kept on ice in KH buffer for isometric tension studies. For protein extraction, the aortic (and other organ) tissue was transferred to KH buffer containing different protease inhibitors (10 μg/mL aprotinin, 5 μg/mL leupeptin, and 7 μg/mL pepstatin). For homogenization, frozen aorta or other organs were pulverized in liquid nitrogen using a mortar and pistil, and the tissue powder was transferred to homogenization buffer consisting of KH buffer containing 1% Triton X and phosphatase/protease inhibitor cocktail (Sigma-Aldrich, Schnelldorf, Germany). After removal of the insoluble parts by centrifugation, the protein concentration was determined by Bradford assay. Tissue samples for mRNA extraction were stored separately at −80 °C, and workup was according to a standard phenol extraction method, followed by RNA quality control using a NanoDrop device (Thermo Fisher Scientific, Langenselbold, Germany).

For cryosectioning, aorta rings of 4 mm length and small parts of frontal cortex from the brain and parts of the heart were used. The different tissue samples were placed into small reservoirs of aluminum and filled with 1 mL of a viscous resin Tissue-Tek^®^ (O.C.T., Tissue-Tek, Sakura, Staufen, Germany). Rapid freezing of the resin and the embedded tissue samples was initiated by placing the reservoirs in liquid nitrogen. Afterwards, the containers were removed from the block of frozen resin. The prepared cryosamples were stored at −80 °C and cut with a cryostat (CM3050S, Leica Biosystems GmbH, Nussloch, Germany) at −25 °C. The resulting cryosections were 8 μm in thickness and were placed on a microscope slide (Thermo Fisher Scientific, Langenselbold, Germany). The slides were stored at −80 °C until further evaluation.

### 2.3. Hemin Treatment

C57BL/6J mice were randomly assigned to four groups: control (CTR), hemin-treated (HEMIN), noise-exposed (NOISE), and hemin-treated plus noise-exposed (HEMIN + NOISE). Noise was applied as a regimen of 4 days of successive exposure. HO-1 induction was carried out by intraperitoneal injection (IP) of hemin (25 mg/kg [17], and stock solution was made freshly in 25 mg/mL dimethyl sulfoxide (DMSO) and diluted 1:3 in PBS) every 2 days during aircraft noise exposure. The time schedule of all treatments is shown in Figure 1.

### 2.4. Dimethyl Fumarate Treatment

C57BL/6J mice were randomly assigned to four groups: control (CTR), DMF-treated (DMF), noise-exposed (NOISE), and DMF-treated plus noise-exposed (DMF + NOISE). DMF is an activator of NRF2 and an inducer of HO-1 and other protective enzymes (e.g., superoxide dismutases and glutathione peroxidase) and was used to examine the NRF2 signal cascade, which encompasses HO-1 response. DMF was dissolved into drinking water at a concentration of 1.6 mg/mL for administration by gavage every day during noise exposure at a dose of 20 mg/kg. The protocol was adopted as previously described [15,18]. The time schedule of all treatments is shown in Figure 1.

### 2.5. Noise Exposure

Mice were continuously exposed to noise (24 h exposure protocol) using repetitive playbacks of a 2 h long noise pattern of 69 aircraft noise events with a duration of 43 s [4,5]. The maximum sound pressure level was 85 dB(A), and the average sound pressure level (SPL) Leq (3) was 72 dB(A), which does not lead to hearing loss. The detailed protocol was previously published [19]. Noise events were interrupted by random silent periods to prevent early adaptation. During the silent periods, the mice were exposed to a background noise level of approximately 50 dB(A). Loudness and corresponding sound pressure levels were adjusted with a class II sound level meter (Casella CEL-246) within one of the cages at initial setup. The SPLs during the exposure procedures were continuously recorded and controlled daily.

### 2.6. Noninvasive Blood Pressure (NIBP)

Noninvasive blood pressure (NIBP) measurements were performed daily throughout the noise exposure regimen (CODA 2, Kent Scientific, Torrington, CT, USA) [4,5]. Baseline measurement was done 1 day before noise exposure started, and then blood pressure was measured daily during the noise exposure regimen (see Figure 1). Animals were placed in restrainers on a preheated plate (32 °C). The CODA System comprises two tail-cuffs for the measurement of blood pressure. An occlusion cuff and a volume–pressure recording cuff (VPR) were placed on the tail of the mice to measure volume–pressure changes in the tail vein upon occlusion and release. Data were acquired by CODA Data Acquisition Software. All measurements of the finally recorded NIBP values were preceded by three training sessions to acclimate the animal to the general procedure before the measurement that was used for the final dataset. The mean values of 10 NIBP readings were used for each animal’s daily value.

### 2.7. Isometric Tension Studies

We evaluated endothelial function through isometric tension studies. This assay records relaxation patterns of aortic ring segments (4 mm) upon subjection to vasodilators and vasoconstrictors in chambers. The rings were preconstricted with prostaglandin F_2α_, resulting in roughly 80% of the maximal KCl-induced tone. Concentration–relaxation curves were performed in response to increasing concentrations of acetylcholine (ACh) and nitroglycerin (GTN) as described [20,21].

### 2.8. Detection of Oxidative Stress and Inflammation in Cortical, Cardiac, and Aortic Tissues

Reactive oxygen species (ROS) formation was determined using dihydroethidium (DHE, 1 μM)-dependent fluorescence microtopography in cryosections of the aorta, frontal cortex from the brain, and the heart as described [22,23]. Cryosections were incubated with DHE for 30 min in PBS. From each animal, three slices were stained and quantified, yielding three pictures per animal that were averaged to produce one mean value (the data point shown in the graphs). ImageJ was used to quantify the images. Data normalization was first based on having slices from all groups on the same object holder (in order to avoid differences in staining from day to day). Second, all fluorescence staining data were normalized to the same tissue area by putting a “box” with a defined area on the stained tissue image as described in detail previously [24]. By this way, we ensure that the fluorescence from each picture is normalized to a specific area.

### 2.9. Detection of Bilirubin by HPLC Method Plasma

A modified HPLC-based method to quantify bilirubin levels was used [25]. Plasma (200–300 µL) was diluted 1:1 with acetonitrile (Honeywell, Germany) and centrifuged at 20,000× *g* for 10 min at 4 °C, and 200 µL of clear supernatant was transferred to a glass vial and put into the HPLC autosampler together with the bilirubin standards (Sigma-Aldrich). Standards of 5 µM bilirubin in H_2_O/acetonitrile (1:1) were used with a total volume of 200 µL. The HPLC system comprised a control unit, two pumps, a mixer, detectors, a degasser, an autosampler (AS-2057 plus) from Jasco Series 2000 (Groß-Umstadt, Germany), and a UV–VIS detector (UV-2077 Plus). The column (C18-Nucleosil 100−3 (125 × 4)) was purchased from Macherey-Nagel, Düren, Germany. The high-pressure gradient was based on the mobile phases 90% acetonitrile and 5 mM citrate buffer, pH 2.2. The gradient conditions were as follows: minute 00:00, 50% A: 50% B; minute 10:00–18:00, 0% A: 100% B; minute 19:00–20:00, 50% A: 50% B. The flow was set to 1 mL/min, and bilirubin was detected by its absorption at 450 nm at a retention time of 15 min and 45 s.

### 2.10. Western Blotting and Dot Blot Analysis

Protein samples were analyzed by Western blot analysis for heme oxygenase-1 (HO-1, mouse monoclonal, 1:10,000, Abcam, Cambridge, MA, USA) and polyclonal rabbit β-actin (1:2500, Sigma-Aldrich) for normalization of loading and transfer. Additionally, we used the following antibodies for dot blot analysis of blood plasma: interleukin-6 (IL-6, rabbit polyclonal, 1:1000, Abcam, Cambridge, MA, USA), 3-nitrotyrosine (3-NT, rabbit polyclonal, 1:1000, Millipore, Burlington, USA), and 4-hydroxynonenal (4-HNE, goat polyclonal, 1:1000, Sigma-Aldrich). Goat anti-mouse and goat anti-rabbit peroxidase-coupled secondary antibodies (1:10,000, Vector Laboratories, CA, USA) were used for the detection of positive bands along with enhanced chemiluminescence (ECL) development [26]. Equal loading of protein samples in dot blot analysis was ensured by Bradford-based determination of protein concentration and loading of 25 µg of heart or plasma protein to the nitrocellulose membrane in each well.

### 2.11. Quantitative Reverse Transcription Real-Time PCR (qRT-PCR)

An amount of 125 ng of total RNA from heart tissue was subjected to quantitative reverse transcription real-time PCR (qRT-PCR) analysis using a QuantiTect Probe RT-PCR kit (Qiagen) as described previously [27]. TaqMan^®^ Gene Expression assays (tested probe and primer sets “off-the-shelf”) for TATA box-binding protein (TBP, Mm_00446973_m1) were used as an endogenous housekeeping gene, to which all other mRNA expressions were normalized. Probes were used as template for subsequent qRT-PCR analysis using specific primers listed below targeting eNOS (Mm_00435204_m1), HO-1 (Mm_00516004_m1), IL-6 (Mm00446190_m1), and cluster of differentiation 68 (CD68, Mm_00839636_m1). In combination with the Taqman Mastermix (buffer, nucleotide, Taq polymerase, reverse transcriptase), this allows “one-step” qRT-PCR. The quantification of relative mRNA expression levels was based on the comparative ΔΔCt method. The expression of all target gene mRNAs in the different treatment groups was expressed relative to that of control (ΔΔCt).

### 2.12. Statistical Analysis

Results are expressed as means ± SEM. We applied two-way ANOVA (with Tukey’s correction for comparison of multiple means) for comparisons of all parameters as the experimental setup contained two variables per group (noise exposure and drug treatment). We used Prism for Windows, version 8.1, GraphPad Software Inc., for statistical analyses. *p*-Values < 0.05 were considered statistically significant, and symbols of significance are explained in the figure legends.

## 3. Results

### 3.1. Effects of Aircraft Noise, Hemin, and DMF on HO-1 Expression and Its Subproduct Bilirubin

The mRNA expression of HO-1 in cardiac tissue was increased in all treatment groups by trend, whereas the most significantly upregulated HO-1 mRNA levels were observed in the HEMIN + NOISE and DMF + NOISE as compared with control mice (Figure 2A). A similar pattern was found in protein expression measurements of HO-1 in heart tissue (Figure 2B) and in HEMIN + NOISE, DMF, and DMF + NOISE groups also in liver tissue (not shown). A trend of increased renal HO-1 protein expression was observed in DMF and DMF + NOISE treatment groups (not shown). Additionally, DMF mice had the highest plasma bilirubin concentrations, with significance against the control and NOISE-only group (Figure 2C). Bilirubin levels in plasma of HEMIN and HEMIN + NOISE mice were elevated by trend.

### 3.2. Effects of Aircraft Noise and HO-1 Inducer and NRF2 Activator on Blood Pressure

To evaluate the effects of the two inducers of HO-1 and NRF2 (hemin and DMF, respectively) on the pathogenesis and progression of hypertension, we exposed mice to aircraft noise over 4 days. Using the CODA System, we measured blood pressure noninvasively. At time point 0, all groups had comparable blood pressure. Noise caused a significantly increased systolic and diastolic blood pressure after 1 day of exposure, which persisted until the final day of the exposure regimen. However, treated mice exposed to noise did not differ significantly from control groups. The systolic as well as diastolic blood pressure of NOISE-only controls was 15–25 mm HG above those of unexposed controls and HEMIN + NOISE and DMF + NOISE groups. The normalizing effect was stable and was observed over the course of the 4-day treatment period (Figure 3).

### 3.3. Vascular Function Is Protected by Induction of HO-1 and Activation of NRF2

Our group previously published that noise induces endothelial dysfunction [4], reflective in a significant attenuation of ACh-induced relaxation of mouse aorta, but not in endothelium-independent vasodilation using GTN. Induction of antioxidant enzymes with both hemin and DMF positively affected vasodilatation with a difference of +9% and +17% maximal endothelium-dependent relaxation against NOISE-only controls (Figure 4). Preconstriction to KCl and prostaglandin F2α (PGF2α) was slightly but significantly increased in the HEMIN and HEMIN + NOISE groups.

### 3.4. Induction of HO-1 and Activation of NRF2 Maintains Inflammatory Parameters at Basal Levels

Inflammation is a well-characterized parameter of hypertension, and accordingly, we observed upregulation in several inflammatory parameters in plasma and heart tissue. Significant decreases due to both treatments were present for interleukin-6 (IL-6) protein expression in plasma (Figure 5A). In addition, a similar pattern was found in gene expression measurements of IL-6 in heart tissue (Figure 5B). CD68 showed decreases in expression between NOISE-only controls and hemin treatment and showed a stable trend of decrease between the DMF + NOISE group and NOISE-only controls (Figure 5C).

### 3.5. Effects of Aircraft Noise and Induction of HO-1 and Activation of NRF2 on ROS Production in Aortic, Cardiac, and Cerebral Tissues

ROS levels were elevated in NOISE-only controls in aortic, cerebral, and cortical tissues, as evidenced by DHE staining of cryosections. ROS levels were decreased in the HEMIN + NOISE and DMF + NOISE groups, at least by trend (Figure 6A–C). We previously established NOX2 as an important source of reactive oxygen species (ROS) in our model of noise exposure [4]. In accordance with this previous observation, we found increased markers of oxidative stress upon noise exposure that were improved by both treatments, at least by trend. In the heart, the NOISE-only group showed a significant increase in nitro-oxidative stress as measured by 3-nitrotyrosine (3-NT)-positive proteins, which was mitigated by treatment with hemin or DMF (Figure 7A). Renal levels of nitrated proteins were also increased by trend in the NOISE-only group, and this trend was not observed in HEMIN + NOISE and DMF + NOISE groups (not shown). In plasma, the NOISE-only group showed a significant increase in lipid peroxidation as measured by 4-hydroxynonenal (4-HNE)-positive proteins, which was mitigated by trend upon treatment with the drugs (Figure 7B). Noise exposure also caused a significant increase in *eNOS* mRNA expression, which was prevented by hemin treatment and partially mitigated in the DMF groups (Figure 7C). Upregulation of eNOS by noise may indicate uncoupling of the enzyme and counter-regulatory increase in its expression in a “rescue” attempt, a phenomenon that was previously described [4].

## 4. Discussion

With the present study, we confirm our previous observations that aircraft noise exposure of mice causes cardiovascular oxidative stress, inflammation, endothelial dysfunction, and hypertension [4,5]. Molecular proof of a central role of superoxide radicals derived from NOX2 was based on the prevention of adverse cardiovascular effects by noise in mice with genetic deficiency in NOX2 (gp91phox^−/−^) [5], and a major role of immune cell infiltration in the vascular tissue of noise-exposed mice was demonstrated by tissue flow cytometry analysis [4,28]. Here, we investigated the protective potential of HO-1 induction and NRF2 activation to prevent noise-induced cardiovascular damage by treating noise-exposed mice with hemin and DMF. In line with previous reports on the antioxidant and anti-inflammatory properties of these compounds, we observed normalization of noise-induced oxidative stress and inflammation, all of which resulted in improved endothelial function and lower blood pressure in noise-exposed mice.

The cellular redox state largely affects the cardiovascular system, and oxidative stress, in the form of overproduction of ROS, is a major trigger of cardiovascular diseases and risk factors like atherosclerosis, hypertension, and diabetes [29,30,31,32]. Endothelial (vascular) dysfunction is an early hallmark of atherosclerosis and future cardiovascular events, [1] and importantly, endothelial dysfunction was also reported in human subjects upon exposure to aircraft or train noise [33,34,35]. The ^•^NO/cGMP signaling pathway can be especially impaired by adverse redox regulation and oxidative stress [36] as exemplified by superoxide-mediated breakdown of nitric oxide, uncoupling/dysregulation of eNOS, oxidative inhibition of soluble guanylyl cyclase (sGC), and ROS-induced endothelin-1 signaling [37], all of which were also observed in noise-exposed mice [4,5]. Noise exposure may further contribute to impaired endothelial function and hypertension by stress hormone signaling [38]. We previously showed that noise exposure induced higher levels of catecholamines and corticosterone in mice, which was associated with more pronounced endothelin-1 signaling [4]. It is worth noting that stress hormone signaling via hypothalamic–pituitary–adrenal axis and sympathetic nervous system activation is also known to cross-activate the renin–angiotensin–aldosterone system [39,40].

Inflammation is another hallmark of cardiovascular mortality and is itself an independent cardiovascular risk factor (reviewed in [1,41]). Cardiovascular mortality was lowered by targeted anti-inflammatory therapy in patients with psoriasis (interleukin (IL)-17/IL-23 axis) [42,43,44], systemic lupus erythematosus (IL-17A signaling) [45], and rheumatoid arthritis (IL-6, tumor necrosis factor (TNF)-α, and IL-17A cascades) [46,47], further reinforcing the integral role of inflammation in cardiovascular disease. The presence of oxidative stress in almost all cardiovascular diseases contributes to endothelial cell activation with facilitated adhesion/infiltration of immune cells, leading to tissue damage. Accordingly, chronic endothelial cell activation may lead to a persistent low-grade inflammatory phenotype of the vasculature as observed in most cardiovascular diseases [41,48]. A molecular proof of this link between oxidative stress and inflammation in the development of hypertension in mice was provided by genetic ablation of myelomonocytic cells, which prevented all adverse effects of angiotensin-II infusion, including increased blood pressure, endothelial dysfunction, vascular oxidative stress, and inflammation. All negative effects were restored by adoptive cell transfer of monocytes from wild-type mice, but monocytes from *Nox2^−/−^* mice failed to do so [49]. As noise exposure aggravated the cardiovascular damage and dysfunction in angiotensin-II-treated mice [28], similar pathomechanisms may come into play in both cardiovascular risk factors. Since kidney dysfunction plays a central role in the development of hypertension and is directly linked to sympathetic hyperactivity [50], our present preliminary observations that noise may cause oxidative kidney damage and DMF treatment may confer protection (e.g., via HO-1 upregulation) warrant further detailed studies on the adverse effects of noise on the kidney.

In light of the described importance of inflammation and oxidative stress in the development of cardiovascular diseases, we explored the antioxidant and anti-inflammatory properties of an HO-1 inducer and an NRF2 activator for the prevention of cardiovascular damage in noise-exposed mice. NRF2 controls not only oxidative stress but also other fundamental physiological and pathophysiological processes, such as inflammation, reperfusion injury, fibrosis, and cancer [51,52,53]. In some aspects, NRF2 can be regarded as an antagonist of NF-κB by shifting the macrophage activity from an inflammatory M1 to anti-inflammatory M2 phenotype and activating regulatory T cells [54]. Classical and alternative activations of macrophages are also redox-regulated, and NRF2 plays an important role in this differential activation [55], which may explain the here observed normalization of CD68 expression levels by NRF2 activation or HO-1 induction. Accordingly, the bardoxolone derivative (RTA) DH404 activated NRF2 and conferred anti-inflammatory effects (via suppression of the transcription factor NF-κB) and normalized ROS formation and endothelial dysfunction in animals with chronic kidney disease or diabetes [56,57,58]. Numerous antioxidant enzymes are regulated by NRF2, such as ferritin, HO-1, glutathione peroxidase-1 (GPX-1), peroxiredoxin-1 (PRX-1), superoxide dismutases (SODs), and thioredoxins (TRXs) (for review, see [59]), and most of them also have important functions in the cardiovascular system. In addition, DMF confers anti-inflammatory effects on endothelial cells by the inhibition of endothelial VEGF receptor 2 expression, which contributes to its antiangiogenic effects [60].

HO-1, the focus of the present study, catalyzes the rate-limiting step in heme degradation by the generation of equimolar concentrations of biliverdin, ferrous free iron, and carbon monoxide [61], followed by biliverdin conversion to bilirubin by the biliverdin reductase and chelation of free iron by ferritin [62]. Thus, the induction of HO-1 is usually paralleled by the upregulation of ferritin, decreasing the free iron levels and preventing Fenton-type reactions. Bilirubin is one of the most powerful endogenous antioxidants and efficiently scavenges peroxynitrite, superoxide, and hydrogen peroxide [25] and is a more potent inhibitor of lipid peroxidation in vitro as compared with vitamin E [63]. In addition, higher serum bilirubin levels show an inverse correlation with the incidence of coronary artery disease [64]. Bilirubin was reported to suppress the activity of vascular NADPH oxidase [65] to confer inhibition of protein kinase C activity [66] and to suppress adverse inflammatory signaling [67], mechanisms that are known to contribute to the development of cardiovascular damage by noise exposure [5]. Importantly, *Hmox1^−/−^* mice displayed upregulated NOX2 protein expression, vascular oxidative stress, markers of inflammation, endothelial dysfunction, and hypertension in response to angiotensin-II [13]. In contrast, pharmacological HO-1 activation improved cardiovascular complications and oxidative stress in diabetic and nitrate-tolerant animals [68,69]. It is worth noting that hemin-mediated protective effects were partially lost in *Hmox1*^−/−^ animals and were enhanced in *Hmox1*-overexpressing animals [70]. Similar opposite effects of hemin in human renal proximal tubule cells, either with pharmacological HO-1 inhibition or with genetic HO-1 overexpression, were observed under cisplatin-induced apoptosis and necrosis [71], all of which support a central role of HO-1 in the protective effects of hemin.

As a major limitation of our study, we would like to mention that we did not use genetic deletion or pharmacological inhibition of HO-1 to provide a molecular proof of HO-1 as a central target of hemin or DMF. In addition, we did not prove that DMF causes activation of NRF2 (e.g., by translocation from the cytosol to the nucleus) but only adopted this NRF2-related protective mechanism of DMF from previous reports [72,73].

## 5. Conclusions

Noise exposure in mice [4,5,28] and men [33,34,35] was demonstrated to induce a pro-oxidative and proinflammatory phenotype associated with cardiovascular damage, endothelial function, and increased blood pressure (Figure 8). As the HO-1 inducer hemin and the NRF2 activator DMF possess potent antioxidant and anti-inflammatory properties and display beneficial cardiovascular effects (Figure 8), these compounds were expected to ameliorate the cardiovascular damage and dysfunction induced by noise exposure. Indeed, with the present study, we could demonstrate for the first time that hemin and DMF treatments prevent the induction of vascular oxidative stress and inflammation as well as endothelial dysfunction and hypertension in noise-exposed mice. Mechanistically, the prevention of NOX2 upregulation in response to noise exposure by hemin and DMF administration may represent a key mechanism by which these drugs confer their beneficial effects. Suppression of monocyte activation and infiltration markers (e.g., downregulated vascular levels of CD68 and IL-6) in noise-exposed mice by hemin and DMF may represent other important mechanisms of the observed protective effects. Although we did not confirm the central role of HO-1 in hemin-mediated protection, previous reports support this assumption. However, it should be noted that NRF2 controls multiple antioxidant and anti-inflammatory pathways (e.g., via above-mentioned other antioxidant genes but also direct effects on endothelial and immune cells independent of NRF2), and accordingly, HO-1 may be only one of several targets that mediate the protective effects of DMF. In light of the recent and future pandemic of noise-induced cardiovascular disease (especially ischemic heart disease) [74,75,76], low-cost and widely applicable drugs (e.g., in the form of nutraceuticals) that potently prevent the major pathomechanisms of noise exposure cardiovascular damage may represent an attractive strategy to lower the burden of environmental diseases.

## Figures and Tables

**Figure 1 antioxidants-10-00625-f001:**
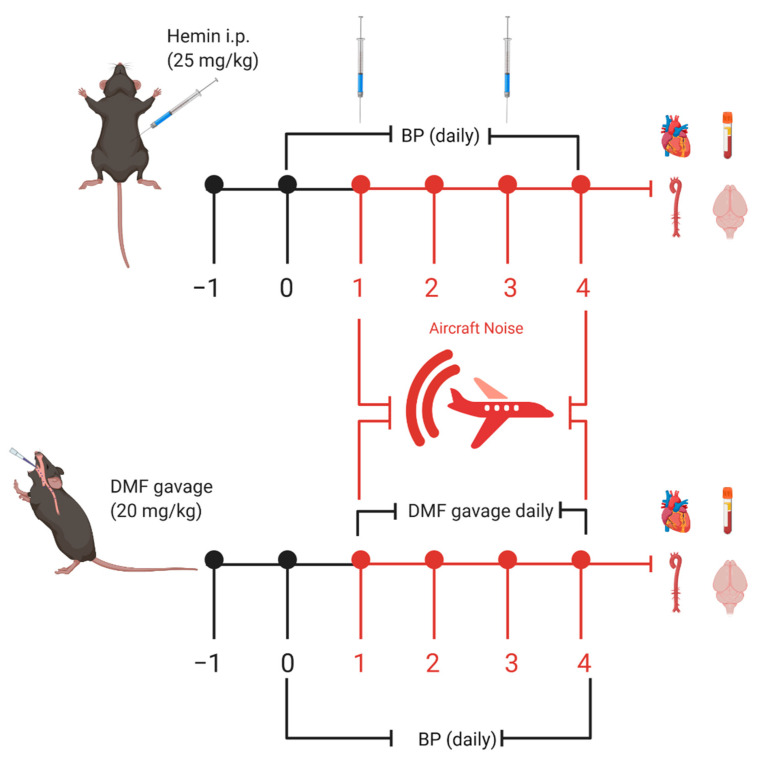
Scheme for treatments and noise exposure. Male C57BL/6J mice were acclimatized to blood pressure measurement, and baseline measurements were taken prior to treatment with hemin or DMF. Hemin was administered once every 2 days during aircraft noise exposure via intraperitoneal injection (IP) injection (25 mg/kg). DMF was administered via daily gavage at a dose of 20 mg/kg. Created with BioRender.com.

**Figure 2 antioxidants-10-00625-f002:**
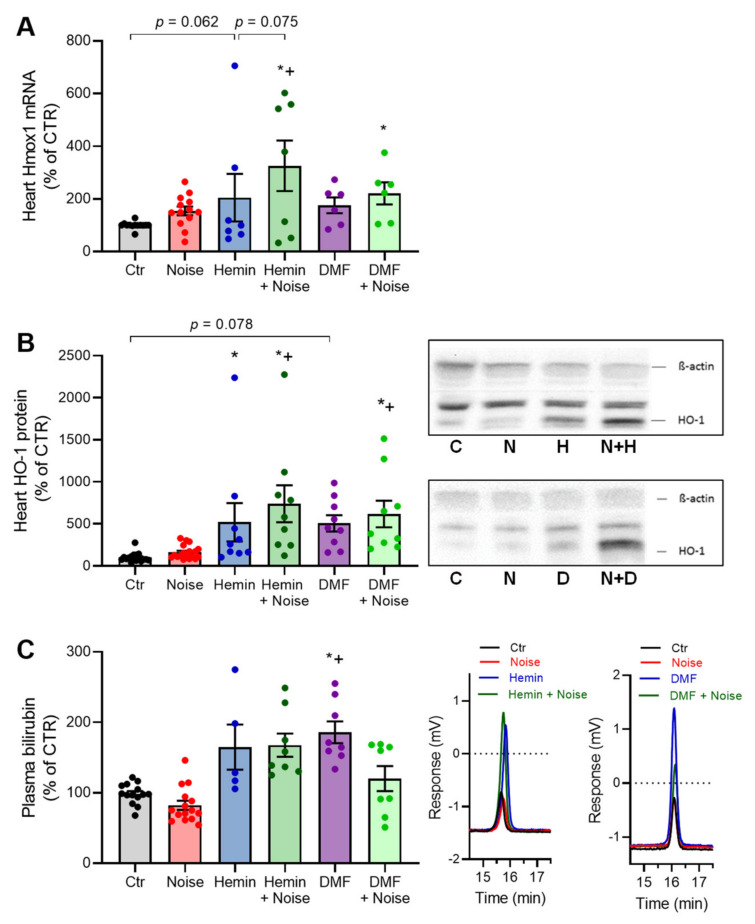
Induction of HO-1 and activation of NRF2 by treatments with hemin or DMF. (**A**) mRNA expression of *Hmox1* in cardiac tissue was measured via quantitative RT-PCR. (**B**) HO-1 protein expression was measured in heart tissue (representative Western blots below the densitometry). (**C**) Quantification and representative chromatograms of bilirubin levels in plasma as measured by HPLC analysis of bilirubin formation and expressed as changes to untreated control. Data points from (**A**) are measurements from 6–13 individual animals, (**B**) represents 9–18 individual samples (each pooled from 2 to 4 mice), and (**C**) represents 5–15 individual samples; * represents *p* < 0.05 vs. untreated control; ^+^ represents *p* < 0.05 vs. + Noise.

**Figure 3 antioxidants-10-00625-f003:**
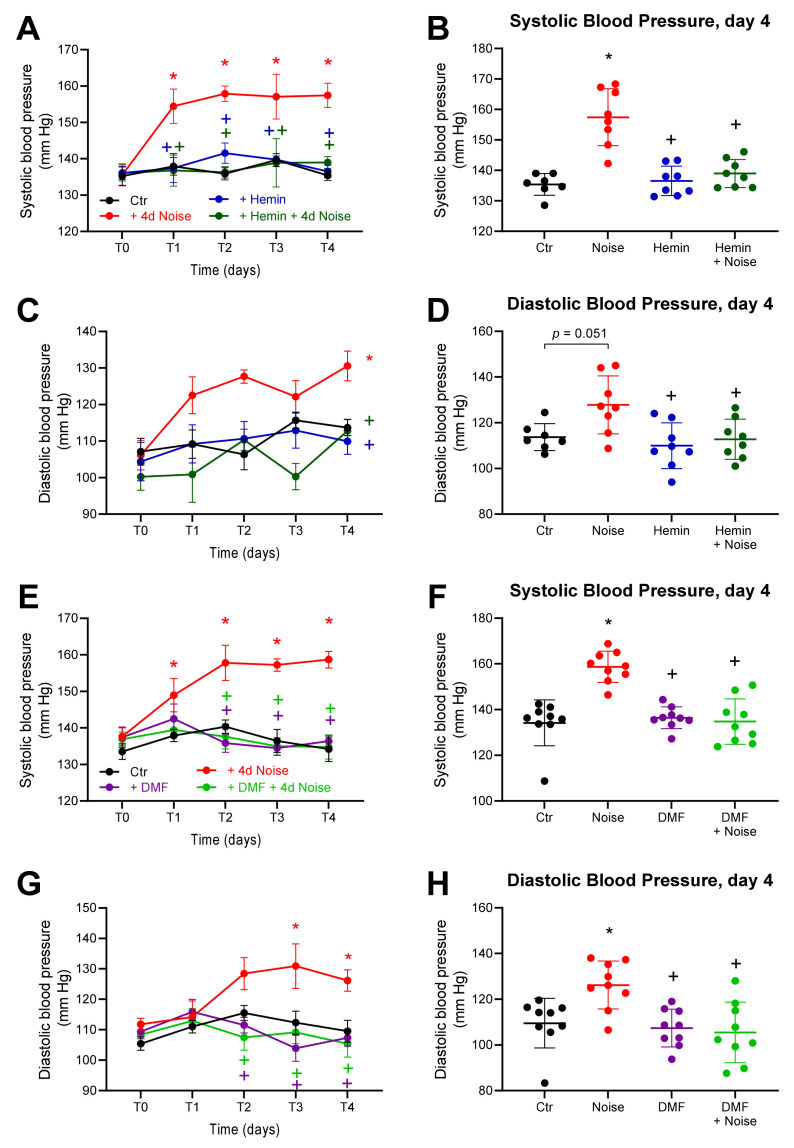
Blood pressure in mice treated with inducer of HO-1 and activator of NRF2 and exposed to noise. (**A**,**E**) The time courses of systolic blood pressure over the span of the hemin and DMF treatments. (**C**,**G**) The respective time courses of diastolic blood pressure over the treatment periods. (**B**,**D**,**F**,**H**) Systolic and diastolic arterial blood pressure measured on the final day of the treatments. Data points are measurements from individual samples; *n* = 8–10. * represents *p* < 0.05 vs. untreated control; ^+^ represents *p* < 0.05 vs. + Noise.

**Figure 4 antioxidants-10-00625-f004:**
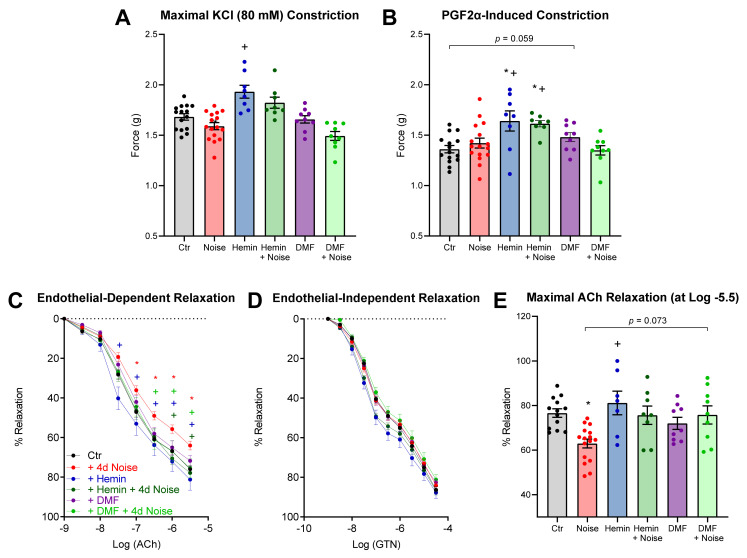
Vascular function in mice treated with inducer of HO-1 and activator of NRF2 and exposed to noise. (**A**,**B**) Potassium chloride (KCl, 80 mM)- or prostaglandin F2α (PGF2α, 3 µM)-induced vasoconstriction. (**C**,**D**) Endothelium-dependent (ACh) and independent (GTN) relaxation of thoracic aortic rings was measured by isometric tension method. (**E**) Quantification of maximum relaxation of all groups. Data points are measurements from individual samples, *n* = 7–16; * represents *p* < 0.05 vs. untreated control; ^+^ represents *p* < 0.05 vs. + Noise.

**Figure 5 antioxidants-10-00625-f005:**
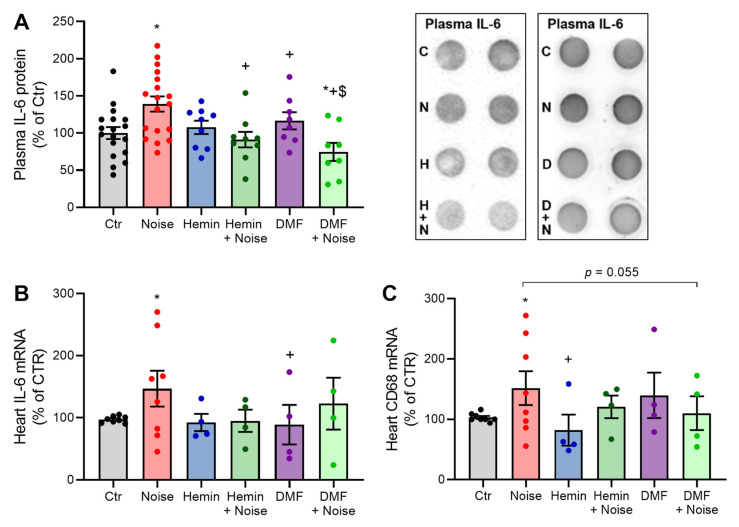
Markers of inflammation show an attenuation in the presence of both treatments. (**A**,**B**) Densitometry and representative dot blot of IL-6-positive proteins in plasma as well as IL-6 mRNA by quantitative RT-PCR in the heart. (**C**) mRNA expression of CD68 in cardiac tissue was measured via quantitative RT-PCR. Data points from (**A**) are measurements from 8–18 individual animals, and (**B**,**C**) represent 4–8 individual samples (each pooled from two mice). * represents *p* < 0.05 vs. untreated controls; ^+^ represents *p* < 0.05 vs. + Noise; ^$^ represents *p* < 0.05 vs. DMF.

**Figure 6 antioxidants-10-00625-f006:**
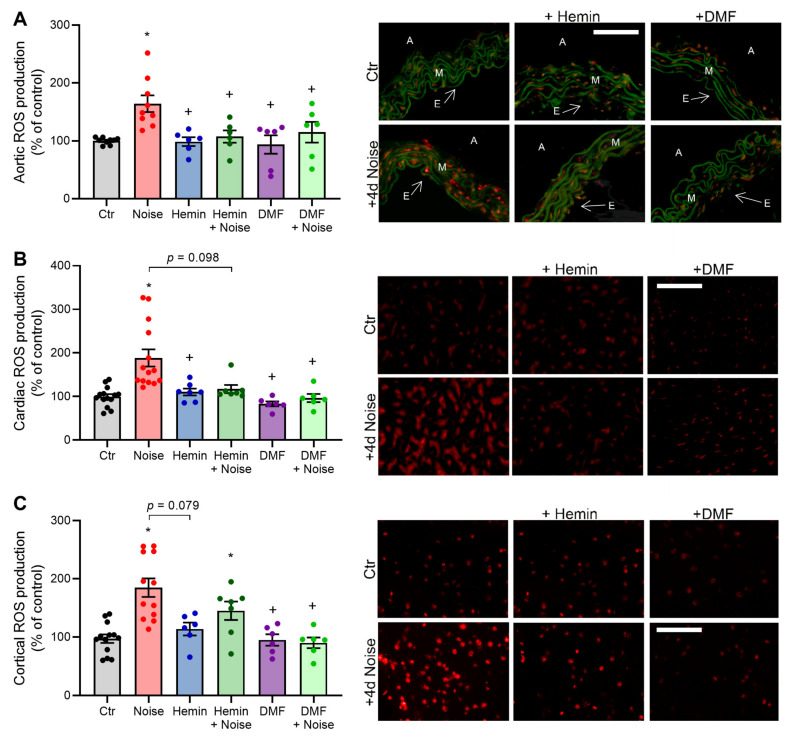
Vascular reactive oxygen species (ROS) formation is decreased by HO-1/NRF2 inducer or activator in noise-exposed mice. (**A**–**C**) Dihydroethidium stainings of aortic, cardiac, and cortical cryosections and their representative photomicrographs show ROS formation as red fluorescence and autofluorescence from aortic laminae as green. A, adventitia; E, endothelium; M, media. Scale bars indicate 100 µm, and a magnification of 20× was used. Data points from (**A**–**C**) are measurements from 6–14 individual animals; * represents *p* < 0.05 vs. untreated controls; ^+^ represents *p* < 0.05 vs. + Noise.

**Figure 7 antioxidants-10-00625-f007:**
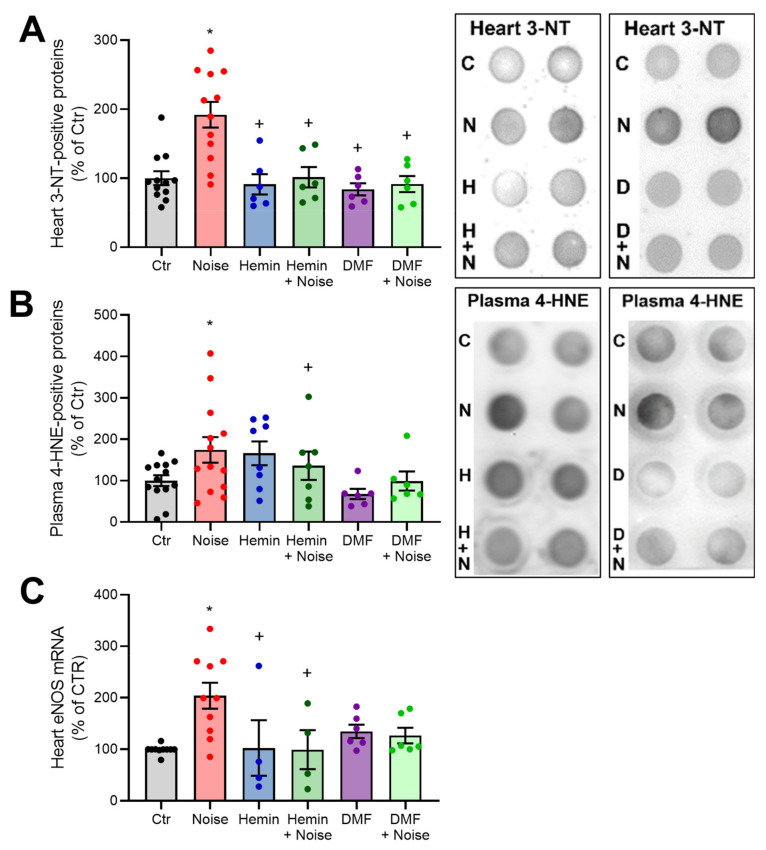
Oxidative stress and sources of ROS are decreased by HO-1/NRF2 inducer or activator in noise-exposed mice. (**A**,**B**) Densitometry and representative dot blots of 3-NT-positive proteins in heart tissue and 4-HNE-positive proteins in plasma. eNOS mRNA via quantitative RT-PCR. Data points from (**A**,**B**) represent 4–13 individual animals, and (**C**) represents 4–10 individual samples, * represents *p* < 0.05 vs. untreated controls; ^+^ represents *p* < 0.05 vs. + Noise.

**Figure 8 antioxidants-10-00625-f008:**
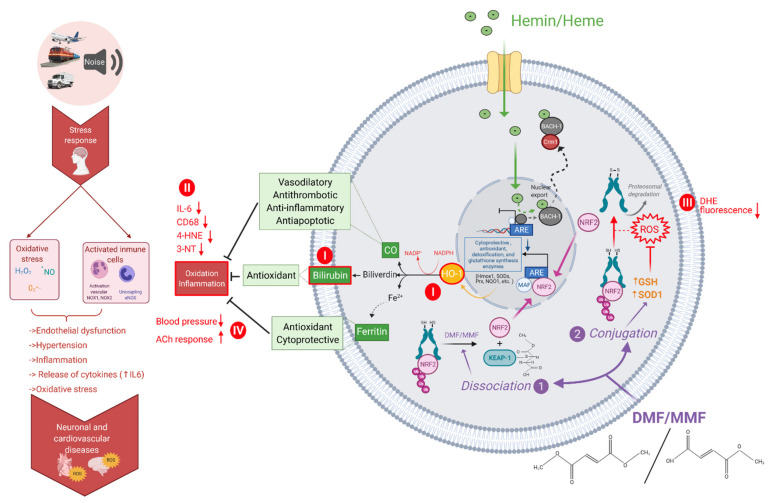
Mechanistic scheme of adverse cardiovascular effects of noise exposure and protection by HO-1/NRF2 induction or activation. Heme or hemin are involved in the dynamic exchange of Bach1 and Nrf2 in the Maf transcription factor network. Enzymatic degradation of heme is catalyzed by HO-1, leading to the formation of biliverdin, carbon monoxide, and ferrous iron. DMF and its primary intestinal metabolite, monomethyl fumarate (MMF), can bind to the cysteine residues of the NRF2/KEAP-1 complex in the cytoplasm or reduce glutathione (GSH) levels. As a consequence, GSH metabolism may affect the oxidative clearance and increase ROS. Besides *Hmox1*, NRF2 also regulates other antioxidant and cytoprotective genes encoding for superoxide dismutases, peroxiredoxins, and others. The effects studied in the present study are marked with red color and roman numbers (**I**,**II**,**III,IV**). We found the upregulation of HO-1 and higher plasma levels of bilirubin in response to hemin and DMF treatments (**I**). Markers of inflammation (IL-6, CD68) and oxidative stress (4-HNE, 3-NT) (**II**) and aortic, cerebral, and cardiac ROS formation (DHE fluorescence) were suppressed (**III**), leading to subsequent improvement of blood pressure and endothelial function (ACh response) (**IV**). Created with BioRender.com.

## Data Availability

The data used to support the findings of this study are available from the corresponding author upon request.

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
