# Peer review of "Noise-Induced Vascular Dysfunction, Oxidative Stress, and Inflammation Are Improved by Pharmacological Modulation of the NRF2/HO-1 Axis"

_antioxidants, 2021, doi:10.3390/antiox10040625_

Round 1

Reviewer 1 Report

The study entitled “Noise-induced vascular dysfunction, oxidative stress and inflammation are improved by pharmacological heme oxygenase-1 induction” presented by Maria Teresa Bayo-Jimenez, shows that pharmacological upregulation of HO-1 is able to counteract noise-induced dysfunction of the aortic endothelium and reduce oxidative stress and inflammation markers in cardiac tissue.

This topic is interesting and definitively merits a closer view.

Major points:

In general, the study design appears appropriate; however, the conclusion, which authors exemplified in the schematic abstract, is, to my opinion, unsupported by data (see below).

  • Authors’ state that both treatments, hemin and dimethyl fumarate (DMF) exert protective effects operating via HO-1. While hemin treatment results in increased HO-1 expression via BACH-repressor release, the mechanism of DMF action is not essentially operating exclusively via HO-1. Without proof, that DMF-mediated protection requires HO-1 induction (genetic modification, or use of pharmacological HO inhibition) this conclusion cannot be drawn. Either way, neither hemin-treatment nor DMF-treatment must necessarily operate via HO-1. Other (indirect) effects cannot be excluded (see also point 2 and 3). Thus, the title is misleading and should be modified accordingly.
  • DMF has profound anti-inflammatory action and on endothelial cells it has been shown to inhibit expression of endothelial VEGF-receptor2, which may explain its anti-angiogenic effect (Meissner-M, PMID: 21430706). DMF is assumed to directly affect endothelial and immune cells, which may operate by other mechanisms, than those involving Nrf2, or one of the downstream targets HO-1. Therefore, the assumed activation of Nrf2, for which experimental evidences have to be included, cannot be put forward as underlying mechanism. Without providing experimental evidences the discussion and the conclusion have to be adapted.
  • The discussion has further to take into account the different sites of action of both pharmacological interventions. The intraperitoneal application of hemin will result in an increased hemin challenge of mainly the liver. Whether or not this route of application will increase HO-1 expression in remote targets, such as the heart, by direct means, remains to be elucidated. Doubtless, continuous hemin application will result in elevated bilirubin levels in the vasculature. The multitude of actions exerted by bilirubin (amongst others the prominent anti-inflammatory effect (PMID: 32082188) can satisfactorily explain the phenomenon observed in this study. A closer consideration must be part of the discussion.

Several issues need further clarification:

  1. The processing of animal organs after sacrifice has to be included into the Material and Method-section. How aortic rings were prepared? How plasma was harvested? Which anticoagulant was used? How slices from heart, brain and aorta were prepared? How RNA and protein was extracted from heart tissue?
  2. Information on how CD68 has been determined is missing.
  3. Please specify how ROS formation was determined? How long slices were incubated? How many slices were evaluated? How data were normalized?
  4. Western blots shown in parts in Fig. 2, and particularly in Fig. 5 and 7 appear very faint. It is problematic that the control protein, beta-actin, is nearly not present, or very weakly stained. Authors should consider the use of another antibody. In the presented blots no clear bands can be discriminated. Additional bands are partially present. Authors should provide the entire blots, without cropping as supplemental material. Loading controls must be included. The dot blots shown in Fig. 5 and Fig. 7 must be shown with loading controls. In the legends these blots require additional information. Also these blots are extremely faint, and given the high heterogeneity of staining, the suitability for quantification is questionable. It appears that the normalizer beta-actin does not work properly, other loading controls (total protein) must be included.
  5. The presented results are not entirely clear. What is displayed in the bar charts – means? Medians? With standard deviation? With standard error of mean? Please indicate.
  6. The statistical analyses indicated raise a certain doubt. Given the high variability of most parameters, two-way ANOVA should be applied everywhere. Additionally, frequently the differences between the “noise” group and those plus treatment were indicated as statistically significant, although the single data points show a very high and overlapping variability. If analyses were paired, such as result could be possible. Or have outliers been excluded? Anyway, the exact number of data points in each single group should be indicated, as it varies within analyses. And it is necessary to increase the font of each indication inside the Figs (the name of the proteins, the signs indicating statistical significance, ect).
  7. When displaying qPCR data, please include background data on the suitability of the PCR assays used. Primer sequences must be indicated. Please adhere to the MIQE guidelines (indicate results of no template controls, RT-minus controls, slope of qPCR using cDNA dilution series ect).
  8. The effects of the pharmacological treatment on other relevant physiological parameters indicating stress – particularly the mentioned hypertension - should be included as supplemental material, such as physical activity and body temperature. Of high value would be the measurement of corticosteroid levels (for instance in feces).

Minor:

Line 59: The format of these lines appears different

Line 62: Abbreviations have to be explained within the plain text, when first mentioned (DMF).

Line 75: Sentence appears incomplete.

Line 76: Please use uniform indications: hemin should not be written with initial capital letter.

Line 91: Since hemin was applied several times, please skip “once”.

Line 111: This sentence is not clear. Please clarify “within one the cages”.

Line 121: Please specify, what is the difference between “training session” and “measurement”? How was the reproducibility within measurements? Why 10 measurements were pooled?

Line 127: Authors write that rings were pre-constricted. Since relaxation may depend on previous constriction, absolute data should be provided (as supplemental material). Have differences been detected between the treatment groups regarding the pre-constriction data?

Line 195: Relevance of Nrf2 activation for the DMF-effect has not been shown. Insinuation of an assumed but not shown mechanism should be avoided; therefore this Fig. title is inappropriate (additionally for Fig. 3, 4, 6, and 7).

Line 220: Symbols for indicating statistical significant differences, which are not applicable in the respective Fig. must be omitted (applies also to the Fig. 4, 5, 6 and 7).

Line 240: Please explain the reason of investigating CD68 at RNA level (in which models this has been established as a suitable marker for inflammation) and indicate how this was achieved. Sequence of primer pairs should be included.

Line 247: Visualization of data should use identical colors and order of groups among figures. Please apply this for the Fig. 5 D.

Line 276: Please indicate what the letters (A, E, M) in the upper panel of graphs stand for.

Author Response

Dear Reviewer,

please find our response in the attached file.

All page numbers refer to the manuscript version with changes marked.

Sincerely

Andreas Daiber

Reviewer 2 Report

This is an interesting study examining the effects of direct HO-1 induction with hemin as well as HO-1 induction via activation of NRF2 on the cardiovascular response to chronic noise stress. Overall, the findings are interesting and supported by the data and the discussion appropriate.

1) It in not clear from the methods if blood pressures were measured before, during or after noise stress. This needs to be clarified.

2) The role of the kidney is very important in the chronic regulation of blood pressure. Can the authors provide data on the effect of hemin and DMF on renal HO-1 levels. Can this be a potential mechanism for the lower blood pressure observed with these treatments?

3) Similarly, do the authors have any data regarding the central effects of HO-1 induction in the current model. Do alterations in central sympathetic outflow play any role in the observed response to hemin or DMF in the present study?

4) Images in figure 6 lack scale bars

5) Do the authors have any data on the production of reactive nitrogen species such as Peroxynitrite in this model. Are they altered by HO-1 induction?

Author Response

Dear Reviewer,

thank you very much for the careful reading of our manuscript and providing valuable comments.

Please find our response in the attached file.

All page numbers refer to the manuscript version with changes marked.

Sincerely

Andreas Daiber

Reviewer 3 Report

This is a nice and important study demonstrating that heme oxygenase-1 has the potential to prevent the harm of severe noise as reflected by hypertension, endothelial dysfunction, inflammatory response, production of reactive oxygen/oxidative stress. Authors employed both Bach1 and RNF2 regulatory pathways to up-regulate HO-1. The experiments are carefully conducted.

Comments

The benefits observed in the study were suggested to be attributed to enzyme activity of HO-1. Indeed, bilirubin generation was demonstrated to be increased in mice treated with heme. The use of HO-1 inhibitors to assess the loss of preventive actions of HO-1 activity against the harm of noise would confirm the conclusion.

Author Response

(The authors gave the same response as above.)

Round 2

Reviewer 1 Report

The revision of the manuscript, now entitled “Noise-induced vascular dysfunction, oxidative stress and inflammation are improved by pharmacological modulation of the NRF2/HO-1 axis”, presented by Maria Teresa Bayo-Jimenez, has significantly improved. The methodological questions have all been answered. Thus data interpretation is now fully comprehensive.

However, some details still need clarification:

  • The Fig 2, which authors have changed, is not entirely clear for me. Please check the new Fig 2B. It is not identical with the former Fig. 2A (see pdf – please also check Y-axis, which shows different values).
  • Please limit the schematic abstract (Fig 8) to those metabolites and networks, which have been investigated in this study (as was suggested in the previous review, in point 1-3). The role of ferritin or CO was not addressed, nor was Nrf2 or the ubiquitination/degradation of Nrf2 an issue. The same applies to SOD/GSH levels. Additionally, these hypothetical details are not necessary to my opinion, because the data obtained in this study already provide a nearly entire picture. Thus, the assumed antioxidative/anti-inflammatory action should be kept more generic for all those metabolites, which were not investigated mechanistically.
  • Please use identical group names: Either B6 or Ctr (better) for the control group in all Figs.

Author Response

The revision of the manuscript, now entitled “Noise-induced vascular dysfunction, oxidative stress and inflammation are improved by pharmacological modulation of the NRF2/HO-1 axis”, presented by Maria Teresa Bayo-Jimenez, has significantly improved. The methodological questions have all been answered. Thus data interpretation is now fully comprehensive.

Answer: We thank the reviewer once more for careful reading and helpful comments as well as for his/her general favorable evaluation. A native speaker has proof-read the MS again and corrected some minor typos and grammar errors.

However, some details still need clarification:

  • The Fig 2, which authors have changed, is not entirely clear for me. Please check the new Fig 2B. It is not identical with the former Fig. 2A (see pdf – please also check Y-axis, which shows different values).

Answer: In response to your previous specific comment #4 we had answered the following in our last point-by-point response: „We reblotted and reevaluated all blots again and selected new representative stainings with a longer exposure time.“ Reblotting was done for all Western blots as well as for all dot bots by another colleague as the one performing the initial blots. Accordingly, not only figure 2A looks slightly different but also 5A, 7A and 7B. Please note that the overall pattern of the data looks similar.

  • Please limit the schematic abstract (Fig 8) to those metabolites and networks, which have been investigated in this study (as was suggested in the previous review, in point 1-3). The role of ferritin or CO was not addressed, nor was Nrf2 or the ubiquitination/degradation of Nrf2 an issue. The same applies to SOD/GSH levels. Additionally, these hypothetical details are not necessary to my opinion, because the data obtained in this study already provide a nearly entire picture. Thus, the assumed antioxidative/anti-inflammatory action should be kept more generic for all those metabolites, which were not investigated mechanistically.

Answer: We think there should be some background information on the potential protective mechanisms of hemin and DMF in the scheme. Also creating the scheme was quite laborious. Therefore, we would like to keep the scheme. However, we see your point and have highlighted the parameters that were investigated in the present study with red color and roman letters I, II, III and IV. These specific findings were also explained in the figure legend.

  • Please use identical group names: Either B6 or Ctr (better) for the control group in all Figs.

Answer: We now used „Ctr“ in all graphs as suggested.